# Muting Bacterial Communication: Evaluation of Prazosin Anti-Quorum Sensing Activities against Gram-Negative Bacteria *Pseudomonas aeruginosa*, *Proteus mirabilis*, and *Serratia marcescens*

**DOI:** 10.3390/biology11091349

**Published:** 2022-09-13

**Authors:** Abrar K. Thabit, Khalid Eljaaly, Ayat Zawawi, Tarek S. Ibrahim, Ahmed G. Eissa, Samar S. Elbaramawi, Wael A. H. Hegazy, Mahmoud A. Elfaky

**Affiliations:** 1Pharmacy Practice Department, Faculty of Pharmacy, King Abdulaziz University, Jeddah 21589, Saudi Arabia; 2Department of Medical Laboratory Sciences, Faculty of Applied Medical Sciences, King Abdulaziz University, Jeddah 21589, Saudi Arabia; 3Vaccines and Immunotherapy Unit, King Fahd Medical Research Center, King Abdulaziz University, Jeddah 21589, Saudi Arabia; 4Department of Pharmaceutical Chemistry, Faculty of Pharmacy, King Abdulaziz University, Jeddah 21589, Saudi Arabia; 5Medicinal Chemistry Department, Faculty of Pharmacy, Zagazig University, Zagazig 44519, Egypt; 6Department of Microbiology and Immunology, Faculty of Pharmacy, Zagazig University, Zagazig 44519, Egypt; 7Pharmacy Program, Department of Pharmaceutical Sciences, Oman College of Health Sciences, Muscat 113, Oman; 8Department of Natural Products, Faculty of Pharmacy, King Abdulaziz University, Jeddah 21589, Saudi Arabia; 9Centre for Artificial Intelligence in Precision Medicines, King Abdulaziz University, Jeddah 21589, Saudi Arabia

**Keywords:** quorum sensing, bacterial virulence, α-adrenoreceptor, biofilm formation, prazosin, antimicrobial resistance

## Abstract

**Simple Summary:**

Bacterial infections are considered one of the main challenges to global health. Bacterial virulence is controlled by interplayed systems to regulate bacterial invasion and infection in host tissues. Quorum sensing (QS) plays a crucial role in regulating virulence factor production, thus could be considered as the bacterial communication system in the bacterial population. The current study aimed to assess the anti-QS and anti-virulence activities of α-adrenoreceptor prazosin against three virulent Gram-negative bacteria. It was demonstrated that prazosin significantly downregulates the expression of QS-encoding genes and shows considered ability to compete on QS proteins in tested strains. Prazosin can significantly diminish biofilm formation and production of virulent enzymes and mitigate the virulence factors of tested strains. However, more testing is required alongside pharmacological and toxicological studies to assure the potential clinical use of prazosin as an adjuvant anti-QS and anti-virulence agent.

**Abstract:**

Quorum sensing (QS) controls the production of several bacterial virulence factors. There is accumulative evidence to support that targeting QS can ensure a significant diminishing of bacterial virulence. Lessening bacterial virulence has been approved as an efficient strategy to overcome the development of antimicrobial resistance. The current study aimed to assess the anti-QS and anti-virulence activities of α-adrenoreceptor prazosin against three virulent Gram-negative bacteria *Pseudomonades aeruginosa*, *Proteus mirabilis*, and *Serratia marcescens*. The evaluation of anti-QS was carried out on a series of in vitro experiments, while the anti-virulence activities of prazosin were tested in an in vivo animal model. The prazosin anti-QS activity was assessed on the production of QS-controlled *Chromobacterium violaceum* pigment violacein and the expression of QS-encoding genes in *P. aeruginosa*. In vitro tests were performed to evaluate the prazosin effects on biofilm formation and production of extracellular enzymes by *P. aeruginosa*, *P. mirabilis*, and *S. marcescens*. A protective assay was conducted to evaluate the in vivo anti-virulence activity of prazosin against *P. aeruginosa*, *P. mirabilis*, and *S. marcescens*. Moreover, precise in silico molecular docking was performed to test the prazosin affinity to different QS receptors. The results revealed that prazosin significantly decreased the production of violacein and the virulent enzymes, protease and hemolysins, in the tested strains. Prazosin significantly diminished biofilm formation in vitro and bacterial virulence in vivo. The prazosin anti-QS activity was proven by its downregulation of QS-encoding genes and its obvious binding affinity to QS receptors. In conclusion, prazosin could be considered an efficient anti-virulence agent to be used as an adjuvant to antibiotics, however, it requires further pharmacological evaluations prior to clinical application.

## 1. Introduction

Bacterial infections constitute one of the major challenges to global health, despite the huge achievements in diagnosis and treatments [1]. What makes it worse, is the ability of bacteria to horizontally gain virulence genes from the same species or even from different species [2,3]. The bacterial virulence expands to include specific bacterial structures such as capsules, pili, and flagella, and production of virulent agents such as enzymes, dyes, and others [4,5], in addition to the formation of biofilms [6,7]. Virulence factors play a key role in the establishment of bacterial infection, guarantee bacterial spreading, escape from the immune system, and even enhance the resistance to antibiotics [6,8]. Bacterial virulence is controlled by interplayed systems to regulate bacterial invasion and accommodation in the host tissues [3,9]. Quorum sensing (QS) plays a crucial role in regulating virulence factor production. Simply, QS can be considered as the bacterial communication system in the bacterial population. Specific QS receptors can sense their cognate inducers that are produced by the same bacterial species or even other species to alter the expression of virulence factors [10,11]. Both Gram-positive and Gram-negative bacteria utilize QS in controlling the virulence, despite their utilization of different inducers and different QS machinery [12,13]. The development of a wide range of virulence factors, such as biofilm formation, bacterial motility, and the synthesis of enzymes including urease, elastase, protease, hemolysins, and other virulent factors, is orchestrated by QS [10,14].

Bacterial development to antimicrobial resistance is an additional complication that worsens bacterial infections and makes treatment difficult, especially in severe infections [1]. Bacteria has the ability to continuously develop resistance to almost all known antibiotics [15]. In the absence of new antibiotics, the innovation of new approaches to fight bacterial resistance is urgently required [16]. One of the innovative approaches is to mitigate bacterial virulence that facilitates the task of the immune system in the eradication of infecting bacteria without affecting bacterial growth so they are not stressed to develop resistance [17,18]. In the light of understanding the QS roles in controlling virulence, targeting QS affords curtailing bacterial virulence [14]. In this context, several chemical moieties and natural products were analyzed for their anti-QS activities that have been proven to have significant effects on bacterial virulence [19,20,21,22,23].

Furthermore, repurposing approved safe drugs was tested as showing efficient anti-QS and anti-virulence agents [24,25,26]. Drug repurposing is a favored strategy because of several advantages, mostly saving time and costs [17].

In a leading study [26], the most clinically prescribed α-adrenoreceptor blockers were in silico screened for their anti-QS activities. The study showed that terazosin and prazosin had the most significant ability to interfere with several QS receptors [26]. Furthermore, terazosin showed significant anti-virulence activities and diminished the virulence of *Pseudomonas aeruginosa* [26] and *Salmonella* Typhimurium [9]. These findings encouraged us to assess the anti-QS and anti-virulence activities of prazosin on Gram-negative bacteria while *Chromobacterium violaceum* as well as, *Pseudomonas aeruginosa*, *Proteus mirabilis*, and *Serratia marcescens* were chosen due to their famed involvement in aggressive infections and significant multidrug resistance profiles. Prazosin works by relaxing the blood vessels so that blood can flow more easily through the body while it is mainly used in treatment of high blood pressure. Prazosin is also useful in treating urinary hesitancy associated with benign prostatic hyperplasia and also effective in improving sleep quality and treating nightmares related to post-traumatic stress disorder [27]. The anti-QS activity of prazosin and its effect on the expression of QS-encoding genes were also evaluated in silico. Moreover, the anti-biofilm and anti-virulence activities of prazosin were evaluated in vitro and in vivo.

## 2. Materials and Methods

### 2.1. Chemicals, Microbiological Media, and Bacterial Strains

Prazosin (CAS number: 19237-84-4) was purchased from Sigma-Aldrich (St. Louis, MO, USA). All media were ordered from Oxoid (Hampshire, UK). The chemicals were of pharmaceutical grade. The bacterial strains *Chromobacterium violaceum* CV026 (ATCC 31532), and *Pseudomonas aeruginosa* PAO1 (ATCC BAA-47-B1), as well as clinical isolates *Serratia marcescens* [21] and *Proteus mirabilis* [28] were fully characterized earlier. 

### 2.2. Determination of Minimum Inhibitory Concentration (MIC) and Effect on Bacterial Growth

The Clinical Laboratory and Standards Institute Guidelines were followed in determining the MICs of prazosin against the tested strains using the broth microdilution technique. The effect of prazosin at sub-MIC (1/4 MIC) on bacterial growth was examined as described earlier [9]. Briefly, the viable count of bacterial cultures with or without prazosin at 1/4 MIC was performed at different time points 4, 16, and 24 h.

### 2.3. Estimation of C. violaceum Violacein Production 

To examine the prazosin anti-QS activity, the effect of prazosin at sub-MIC on the QS-controlled *C. violaceum* violacein. Luria-Bertani (LB) agar plates provided with the autoinducer N-hexanoyl homoserine lactone (C6HSL) were seeded with *C. violaceum*. Prazosin at sub-MIC was added to wells made in the agar plates. The white or cream color formed around the well indicated QS inhibition, while a clear halo indicated antimicrobial activity [29].

The prazosin effect on the production of violacein was quantified as previously described [9,30]. Briefly, LB broth aliquots provided with C6HSL in the absence or presence of prazosin at-sub-MIC were added to equal volumes of *C. violaceum* suspensions (O.D600 1) at room temperature for 24 h. The violacein pigment was extracted with dimethyl sulfoxide (DMSO) and the absorbances were measured at 590 nm. 

### 2.4. Anti-Biofilm Activity Evaluation

Strong biofilm-forming bacterial strains *P. aeruginosa*, *P. mirabilis*, and *S. marcescens* [18] were used to evaluate the prazosin anti-biofilm activity. The crystal violet method was employed to evaluate anti-biofilm activity, and the absorbances of extracted crystal violet at 600 nm [31]. Briefly, bacterial suspensions of tested strains were prepared from fresh overnight cultures and their optical densities were adjusted to OD600 of 0.4 (1 × 10^8^ CFU/mL). Ten μL aliquots of prepared suspensions were mixed with 1 mL of fresh Tryptic soy broth (TSB) in the presence or absence of prazosin at sub-MIC. TSB aliquots (100 μL) with and without prazosin were transferred into the wells of 96-wells microtiter plate and incubated at 37 °C for 24 h. The planktonic cells were aspirated and the wells washed several times with distilled water and left to air dry. The adhered cells were fixed with methanol for 25 min and stained with crystal violet (1%) for 25 min. The wells were washed, the attached dye eluted by 33% glacial acetic acid, and the absorbance measured. The prazosin anti-biofilm activity was visualized by allowing the formation of bacterial biofilms on cover slips in the presence or absence of prazosin, as described earlier [21]. The same procedure as described above was followed except that the biofilms of the tested strains were formed on glass slides placed in polystyrene petri plates in the presence and absence of prazosin at sub-MIC. Light microscope images were captured for the formed biofilms in the presence or absence of prazosin using a Leica DM750 HD digital microscope (Mannheim, Germany).

### 2.5. Protease Production Evaluation

The prazosin effect on protease production by the tested strains was evaluated by applying the skim milk agar method as previously mentioned [18]. Briefly, the supernatants from bacterial cultures with or without prazosin at sub-MIC were obtained. The diameters of the clear zones were measured in millimeters after adding 50 µL to the wells made in 5% skim milk agar plates and incubating them for 24 h at 37 °C.

### 2.6. Hemolysin Production Evaluation

As previously mentioned, prazosin’s anti-hemolytic action was evaluated. Briefly, the supernatants from bacterial cultures grown with or without prazosin at sub-MIC were obtained. Fresh 2% rabbit blood suspensions were combined evenly with the supernatants. The mixtures were centrifuged and the absorbances at 540 nm were measured after a two-hour incubation at 37 °C. Blood suspensions were treated with sodium dodecyl sulfate (SDS) (0.1%) to provide positive and negative controls of hemolyzed blood.

### 2.7. In Vivo Anti-Virulence Evaluation

The in vivo anti-virulence effect of prazosin was assessed using the mice survival model, as previously described [25,26]. Briefly, sub-MIC (1 × 10^6^ CFU/mL) fresh overnight bacterial cultures in LB broth with or without prazosin in phosphate-buffered saline (PBS) were used. Three-week-old female Mus musculus mice were divided into four groups (n = 10). One positive control group received an intraperitoneal (ip) injection of bacterial cultures not treated with prazosin, two negative control groups received either no injections or injections of sterile PBS, and the test group received an intraperitoneal injection of 100 µL of prazosin-treated bacterial cultures. The Kaplan–Meier method was employed to plot and record the mice survival over 5 days.

### 2.8. Real-Time Quantitative Reverse Transcription PCR (RTq-PCR) for QS-Encoding Genes

The RNA of prazosin (at sub-MIC) treated or untreated *P. aeruginosa* was extracted as previously described (Table 1) [26]. The primers used to amplify the QS-encoding genes *lasI*, *lasR*, *rhlI*, *rhlR*, *pqsA*, and *pqsR*, were indicated previously. The extracted RNA was used to synthesize cDNA, and RTq-PCR was performed to evaluate the QS-encoding gene expression, and the relative expression was evaluated by the comparative threshold cycle (2^−∆∆Ct^) method [32].

### 2.9. In Silico Study

PubChem database (https://pubchem.ncbi.nlm.nih.gov/; accessed on 4 May 2022) was used for retrieving the prazosin SMILES string. Over drug repurposing, the compound went through the complete consequence of the drug development cycle [33]. SWISSADME tool (https://www.expasy.org/resources/swissadme; accessed on 4 May 2022) was used for analysis of the prazosin for molecular properties. The compound energy was minimized to 0.1 Kcal/mol/Å^2^ gradient RMS on Molecular Operating Environment (MOE 2019.012) for the docking process.

Molecular Operating Environment (MOE) 2019.0102 was used for the molecular docking processes on the targeted bacterial QS proteins. *P. aeruginosa* QS control repressor (PDB ID: 3SZT) and *P. mirabilis* adhesion MrpH (PDB ID: 6Y4F) crystal structures were downloaded from the RCSB Protein Data Bank (https://www.rcsb.org/; accessed on 4 May 2022). *S. marcescens* QS transcriptional regulator SmaR (Uniprot Entry: Q14RS3) has no resolved crystal structure, so a SWISS MODEL (https://www.expasy.org/resources/swiss-model; accessed on 6 May 2022) was used and active site architecture analyzed for its validation. Protein structures were prepared by using the MOE QuickPrep protocol. The active pocket was validated by re-docking of the co-crystallized ligand and measuring the root-mean square deviation (RMSD) values. Prazosin was docked into the protein active site through Alpha triangle placement through Amber10:EHT force-field using two stages; rigid-receptor protocol and induced-fit protocol. Furthermore, the Computed Atlas for Surface Topography of Proteins (CASTp; http://sts.bioe.uic.edu/castp/index.html; accessed on 8 May 2022) server was used for the active pocket prediction [34].

### 2.10. Statistical Analysis

The findings of the experiments, which were carried out in triplicate, are shown as means ± standard error. Except where otherwise noted, statistical significance was determined by the student’s *t*-test, with a significance when *p* value < 0.05 (GraphPad Prism Software, v.8, San Diego, CA, USA).

## 3. Results

### 3.1. Determination of Minimum Inhibitory Concentration (MIC) of Prazosin and Its Effect on Bacterial Growth

The prazosin concentrations that inhibited the *C. violaceum*, *P. aeruginosa*, *P. mirabilis*, and *S. marcescens* growth were 2, 2, 1, and 1 mg/mL, respectively. The anti-QS and anti-virulence effects of prazosin were evaluated at sub-MIC (1/4 MIC) to rule out any impact on bacterial growth. The tested bacterial strains were grown in the presence of prazosin at a sub-MIC (1/4 MIC) level and their growth was compared to that of the strains grown in the absence of prazosin to demonstrate that prazosin had no effect on bacterial growth. The bacterial counts in the presence or absence of prazosin at various time periods did not change significantly (Figure 1). 

### 3.2. Prazosin Decreased the Production of QS-Controlled C. violaceum Violacein

Prazosin was examined for its action on the biosensor *C. violaceum* synthesis of the QS-controlled pigment violacein in order to do a preliminary evaluation of its anti-QS activity. White (creamy) zones were formed around the wells containing prazosin at sub-MIC indicate QS inhibition (Figure 2A). Furthermore, the effect of prazosin at sub-MIC on violacein production was quantified by comparing the pigment production in its presence or absence spectrophotometrically. Prazosin significantly reduced the production of QS-controlled violacein (Figure 2B). 

### 3.3. Prazosin Showed Significant Anti-Biofilm Activities 

The production of biofilm in bacterial cultures treated or untreated with prazosin at sub-MIC was evaluated using the crystal violet technique. Interestingly, this work shows that prazosin significantly diminished the biofilm formation by the tested strains (Figure 3).

### 3.4. Prazosin Decreased the Production of Virulence Factors 

The effects of prazosin at sub-MIC on the production of protease and hemolysin by *P. aeruginosa*, *P. mirabilis*, and *S. marcescens* were assayed. Prazosin significantly decreased the production of protease and hemolysin (Figure 4).

### 3.5. Prazosin In Vivo Mitigates Bacterial Virulence

The anti-virulence efficacy of prazosin at sub-MIC against *P. aeruginosa*, *P. mirabilis*, and *S. marcescens* was assessed using an in vivo protection assay. All mice survived in the negative control groups injected with sterile PBS or un-injected. On the other hand, only 2 out of 10, 4 out of 10, and 5 out of 10 mice survived when injected with *P. aeruginosa*, *P. mirabilis*, and *S. marcescens*, respectively. Prazosin protected 5 out of 10, 8 out of 10, and 8 out of 10 mice against *P. aeruginosa*, *P. mirabilis*, and *S. marcescens*, respectively. The mice death was recorded by the Kaplan–Meier method and the log-rank test was applied to test the significance. Prazosin significantly reduced the *P. aeruginosa*, *P. mirabilis*, and *S. marcescens* capacity to kill mice, the log rank test for trend (*p* = 0.0009, =0.0261, and =0.0267, respectively) (Figure 5). 

### 3.6. Prazosin Downregulated the Expression of P. aeruginosa QS-Encoding Genes

In control untreated *P. aeruginosa* and in *P. aeruginosa* treated with prazosin (at sub-MIC), the expressions of the *P. aeruginosa* QS-encoding genes LasI/R, RhlI/R, and PQS were assessed. When compared to the control untreated culture, the expression levels of the genes encoding the autoinducers *lasI*, *rhlI*, and *pqsA* and their receptors *lasR, rhlR*, and *pqsR* were markedly downregulated (Figure 6). 

### 3.7. Prazosin Binding Affinity to QS Receptors 

#### 3.7.1. Prazosin Molecular Descriptors and ADME Properties

Prazosin, as a marketed drug, is a suitable target for repurposing. Figure 7 generated by the SwissADME tool shows that prazosin owes suitable physicochemical properties for oral bioavailability [33]. 

The compound possesses molecular weight below 500, hydrogen bond donor < 5, hydrogen bond acceptor < 10, and logP value < 5, rotatable bond count < 10 and topological polar surface area (TPSA) < 140 Å². Hence, the compound obeys Lipinski’s rule of five and Veber’s rule. The PubChem CID, 3D structures, Lipinski’s rule of five rule, and Veber’s rule are shown in Table 2.

#### 3.7.2. Inter-Species Docking Analysis on QS Biotargets

To further investigate the interaction of prazosin with quorum sensing mechanisms, a validated two-stage docking simulation was performed. The first stage is a rigid receptor protocol, and the second stage is an induced-fit protocol. The in silico molecular approach allowed the reliable study of the potential interactions between prazosin with three bacterial targets regulating virulence genes of three different bacteria. The *P. aeruginosa* quorum-sensing control repressor (PDB ID: 3SZT) and the *P. mirabilis* adhesin MrpH (PDB ID: 6Y4F), along with a SWISS model for the *S. marcescens* SmaR (Uniprot ID: Q14RS3) represented the targets for this in silico study. 

The *P. aeruginosa* quorum-sensing transcription factor (QscR) was co-crystallized with *N*- 3′oxo-octanoyl-L-homoserine lactone (3OC12-HSL, strong activating ligand) as homodimer at 2.55 Å resolution. The general topology of the QscR exhibited an α/β/α sandwich. The ligand-binding domain (LBD) at the N-terminus and the DNA-binding domain (DBD) located at the C-terminus are also connected by a short linker of residues from 165 to 174 [35].

The *P. mirabilis* adhesin MrpH structure was solved at 1.75 Å. The architecture demonstrated seven β-strands and two α-helices. The N-terminus is located relatively close to the C-terminus. Cys 128 and Cys 152 form a disulfide bond which is important for maintaining structural integrity and function. The protein structure possesses a tetrahedrally coordinated Zn ion bound by three histidines (His 72, His 74, His 117) and by an external ligand [36].

*S. marcescens* Quorum-sensing transcriptional regulator (SmaR) (Uniprot Entry: Q14RS3, www.uniprot.org; accessed on 4 May 2022) was identified as a target sequence. *C. violaceum* CviR transcriptional regulator (PDB: 3QP5, 3.25 Å) served as a template [37], as predicted using SWISS model (https://www.expasy.org/resources/swiss-model, accessed on 25 May 2022) [38,39,40,41,42]. *C. violaceum* CviR transcriptional regulator is the top-ranked template according to the sequence identity from seven suggesting templates. The binding site in the SWISS model was located by the site finder module in the MOE which matched the co-crystallised ligand (HLC) binding site of the template. Moreover, CASTp was used to verify the binding pocket. 

The Site Finder module is used to find potential 3D pockets that have potential as active sites for the protein. Figure 8 shows the protein structure and putative pockets of the binding site topology at the five bacterial targets. The calculated Richard’s solvent accessible surface area and volume were estimated as; 381.528 Å^2^/681.245 Å^3^, 23.620 Å^2^/3.608 Å^3^, and 664.317/358.797 Å^2^/Å^3^ for the binding sites of *P. aeruginosa* QscR (PDB 3SZT), *P. mirabilis* adhesin MrpH (PDB: 6Y4F), and *S. marcescens* SmaR model, respectively using the Computed Atlas for Surface Topography of Proteins (CASTp). CASTp is built on recent theoretical and algorithmic results of computational geometry. The pockets and cavities are identified analytically, the boundary between the bulk solvent and the pocket is defined precisely and the parameters are calculated rotationally.

#### 3.7.3. Docking Simulations on *P. aeruginosa* QS Control Repressor

Docking simulations of prazosin with QScR (PDB ID: 3SZT) showed comparable results to the docking of the co-crystallized ligand (3OC12-HSL, strong activating ligand) in terms of 3D fitting and filling of the same space in the binding site as shown in Figure 9A, also in terms of hydrophobic interactions with Ser38, Phe54, Tyr58, Tyr66, Val78, and Met127 Figure 9B. 

**Figure 9 biology-11-01349-f009:**
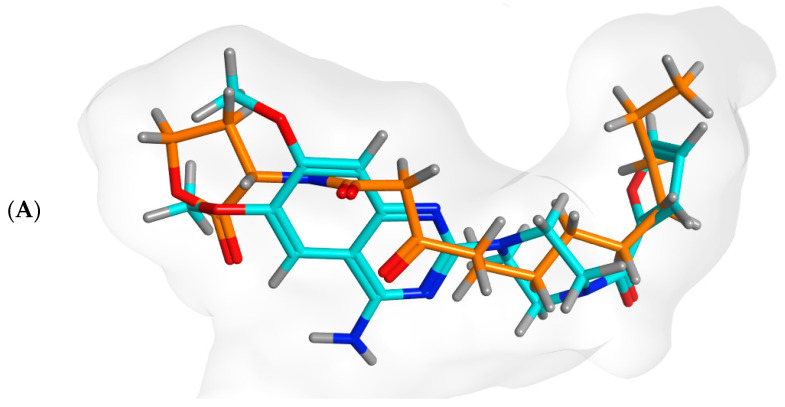
Arg42 and Phe54. However, the co-crystallized ligand was stabilized with different hydrogen bonds with Tyr58, Trp62, and Asp75. These differential binding modes resulted in a slight difference in score S = −9.9177 and −10.1774 for prazosin and 3OC12-HSL, respectively Table 3.

**Table 3 biology-11-01349-t003:** Docking results for both 3OC12-HSL and prazosin with *P. aeruginosa* QS control repressor.

Ligand	Rigid Receptor Protocol		Induced-Fit Protocol		H-bond Interactions	Hydrophobic Interactions	*pi*-Interactions
	S score Kcal/mol	RMSD	S score Kcal/mol	RMSD			
**Prazosin**	**−9.5648**	**1.4357**	−9.9177	1.2806	-	Ser38, Phe54, Tyr58, Tyr66, Val78, Met127	Ser38(*pi*-H), Arg42 (*pi*-H), Phe54(H-*pi*)
**3OC12-HSL**	−10.4256	1.2495	−10.1774	1.1676	Tyr58, Trp62, Asp75	Ser38, Tyr52, Phe54, Tyr58, Tyr66, Val78, Met127	-

#### 3.7.4. Docking Simulations on *P. Mirabilis* Adhesin MrpH

Docking of prazosin with *P. mirabilis* adhesin MrpH revealed the ability of prazosin to act as the external ligand for Zn^+2^ binding, which is crucial for biofilm formation, Figure 10. This is in addition to the comparable hydrophobic interactions with glutamate (co-crystallized ligand). 

The docking results summarized in Table 4 showed the inferiority of prazosin in terms of hydrogen bond formation that was reflected in glutamate having a better score = −8.4332 when compared to −6.0361 achieved by prazosin. However, this may be attributed to the much smaller size of glutamate that enables the molecule to move and bind more freely than a much bigger molecule like prazosin as demonstrated by the superimposition of both molecules in the MrpH pocket Figure 11.

#### 3.7.5. Docking Simulations on *S. marcescens* SmaR Model

4-(4-chlorophenoxy)-*N*-[(3S)-2-oxotetrahydrofuran-3-yl] butanamide (HLC) as the co-crystallized ligand of the template; C. violaceum CviR transcriptional regulator, was docked in the active site for the SWISS model of *S. marcescens* SmaR for its validation. Docking HLC identified key amino acid residues with reasonable RMSD (1.3923 Å) and good binding affinity where the S score was −7.1761 Kcal/mol. Carbonyl group forms H- bonding interaction with Trp53 and hydrophobic contacts with Ala32, Phe44, Tyr57, Trp81, Ile105, Val122, Ser124 amino acid residues Figure 12.

Prazosin showed a good ability to bind to *S. marcescens* SmaR protein, forming two *pi*-H bond interactions with Leu69 and quinazolinyl moiety in addition to hydrophobic interactions with Ala32, Phe44, Tyr57, Leu69, Trp81, Ile105, Val122, and Ser124 Figure 13. The energy docking score for prazosin and HLC (co-crystallized ligand) Table 5. This indicates the prazosin ability to antagonize receptor function, and this might result in the inhibition of QS and its regulated virulence factors. 

## 4. Discussion

QS amends bacterial virulence via controlling the expression of virulence factors’ encoding genes [25]. Mitigating bacterial virulence controlled by QS is an interesting approach that may be applied to conquer the overstated pathogenesis and resistance development of antibiotics [43]. Despite the considered successes of antibiotics that have been achieved since their first discovery, bacteria have managed to develop resistance to almost all antibiotic classes [1,16]. It is well known that affecting bacterial growth could stress them into developing resistance to the used antibiotics [15]. The main advantage of targeting QS systems is curtailing the bacterial virulence to ease the immune system’s task in facilitating eradication of bacteria without affecting bacterial growth, and hence avoiding stressing bacteria to develop resistance [31,44]. In a previous study, α-adrenoreceptor blockers were in silico screened for their anti-QS activity that sheds light on the considered ability of prazosin to compete on different QS receptors [26]. 

The main concept of targeting QS is to affect bacterial virulence without affecting growth to avoid resistance development [18,45]. In this context, the prazosin was tested at a sub-MIC concentration to avoid any effect on bacterial growth. Prazosin at sub-MIC did not significantly influence the growth of the tested strains. Our previous in silico findings showed prazosin’s ability to bind to LuxR-type QS receptor *C. violaceum* CviR (PDB: 3QP5) [26]. *C. violaceum* is a Gram-negative bacterium and produces its violet pigment violacein in response to QS regulation of violacein-encoding genes *vioA-D* [30,46]. *C. violaceum* biosensor strain CV026A is a mutant of the wild type lacking autoinducer synthase and requires exogenous autoinducer (N-acyl homoserine lactones) to release violacein [47]. *C. violaceum* CV026A confers a suitable tool for screening QS activities, and it has been used routinely to evaluate QS activity in Gram-negative bacteria [9,25,47]. Prazosin at sub-MIC significantly reduced the production of violacein, which indicates the possible prazosin anti-QS activities. 

*P. aeruginosa* is one of the most important human pathogens that can cause serious systemic infections, such as eye, burn, wound, respiratory tract, nosocomial, and blood infections [20]. *P. aeruginosa* could acquire resistance to nearly all classes of antibiotics which complicates its treatment [48]. *P. aeruginosa* utilizes several interplayed secretion and QS systems to orchestrate the production of its huge arsenal of virulence factors [30,49]. *P. aeruginosa* employs mainly three types of QS systems namely Lux-types including RhlI/R and LasI/R, and non-Lux type PQS in addition to Lux analogues QscR [35]. *P. mirabilis* causes serious infections, such as nosocomial, urinary tract, wound, and burn infections [50]. The *P. mirabilis* virulence is owing to its inherent capability of peritrichous flagellar motility, formation of biofilm and enzyme production, such as urease, protease, and hemolysins which facilitate the infection spread [28]. *P. mirabilis* also demonstrates increased multi-drug resistance profiles [50]. *S. marcescens* is one of the most frequent nosocomial pathogens, causing pneumonia and blood infections [21,51]. *S*. *marcescens* QS modulates biofilm formation, swarming and sliding motilities, hemolytic activity, and production of biosurfactant and enzymes such as protease, lipase, nuclease, and chitinase [18,52,53]. Two main QS systems have been identified in *Serratia* spp. SwrI/R and SmaI/R [53]. It is worthy of mention that both *P. mirabilis* [28], and *S*. *marcescens* used in this study were isolated from macerated diabetic foot wound and endotracheal aspiration, respectively, and demonstrate multi-drug resistant profiles. 

Bacteria biofilms are membrane attached communities in which polymer matrixes mainly composed of polysaccharides are produced to hold bacteria together [6,20]. Bacteria resist environmental stress by formation of biofilms in which the bacterial resistance to antibiotics is augmented in comparison to planktonic bacterial cells. So, the bacterial biofilm formation constitutes an additional obstacle, and their eradication from animate or inanimate objects is an essential requirement for efficient antibiotic treatments [6]. QS plays the crucial role in the formation of bacterial biofilms and targeting QS results in diminishing their formation [43]. The tested strains *P. aeruginosa*, *P. mirabilis* [28], and *S. marcescens* [52] were documented as strong-biofilm forming. Interestingly, prazosin significantly diminished the biofilm formation in the three tested strains. 

QS modulates the production of diverse virulence extracellular enzymes like lipase, protease, elastase, hemolysins, urease, and other virulence factors [14]. The production of these enzymes facilitates the spread and establishment of bacterial infections into host tissues [21,54,55]. For instance, proteolytic and hemolytic activities significantly enhance bacterial pathogenesis [54,55]. Prazosin significantly decreased *P. aeruginosa*, *P. mirabilis*, and *S. marcescens* production of hemolysins and proteases. In agreement with the in vitro phenotypic prazosin anti-virulence activities; prazosin at sub-MIC showed a significant reduction in the killing capacity of *P. aeruginosa*, *P. mirabilis*, and *S. marcescens*, protecting the tested mice. 

Prazosin significantly downregulated the expression of *P. aeruginosa* QS receptors RhlR, LasR, and PqsR encoding genes and their synthetase encoding genes *rhlI*, *lasR*, and *pqsA*, respectively. Our detailed molecular docking work was performed to evaluate the prazosin binding affinity to the *P. aeruginosa* QscR, *P. mirabilis* MrpH, and *S. marcescens* SmaR. *P. aeruginosa* QscR is a Lux analogues QS receptor that does not have a partner LuxI homolog but can sense LasI autoinducers [53]. *S. marcescens* SmaR is Lux type QS receptor that senses C4- and C6-homoserine lactone to control the expression of virulence genes encoding lipase, nuclease, protease, hemolysin, prodigiosin, and biofilm formation [21,53]. Mannose resistant proteus (MRP) like fimbriae are important for biofilm formation, aggregation, and colonization of *P. mirabilis*, in bladder and kidney [56]. The Zn-dependent receptor-binding domain of *P. mirabilis* adhesin MrpH plays a key role in adhesion and biofilm formation and is considered one of the QS proteins [28,36]. In agreement with in vitro and in vivo findings, the docking study showed a considered ability of prazosin to compete on different QS proteins showing possible anti-QS activity. 

The current findings show the considered prazosin anti-QS and anti-virulence activity that can potentially be used alone or as adjuvant to antibiotics. Moreover, this study hints at the use of prazosin as a pharmacophore to synthesis new active compounds to be used for anti-virulence. Although, this study suggests the anti-virulence activities of prazosin, larger multicenter studies alongside pharmacological, toxicological, and pharmaceutical studies are very much needed to further investigate its potential use in clinical settings. 

## 5. Conclusions

Mitigation of bacterial virulence is a promising approach to overcoming bacterial resistance development. QS controls the regulation of several virulence factors and hence interfering with QS could guarantee diminishing bacterial virulence. The present study demonstrates that prazosin significantly downregulated the QS-encoding genes and shows considered ability to compete on QS proteins in *P. aeruginosa*, *P. mirabilis*, and *S. marcescens*. Prazosin can significantly diminish the formation of biofilm, virulent enzyme production and mitigate the virulence factors of *P. aeruginosa*, *P. mirabilis*, and *S. marcescens*. However, more testing is required alongside pharmacological and toxicological studies to assure the potential clinical use of prazosin as an adjuvant anti-QS and anti-virulence agent.

## Figures and Tables

**Figure 1 biology-11-01349-f001:**
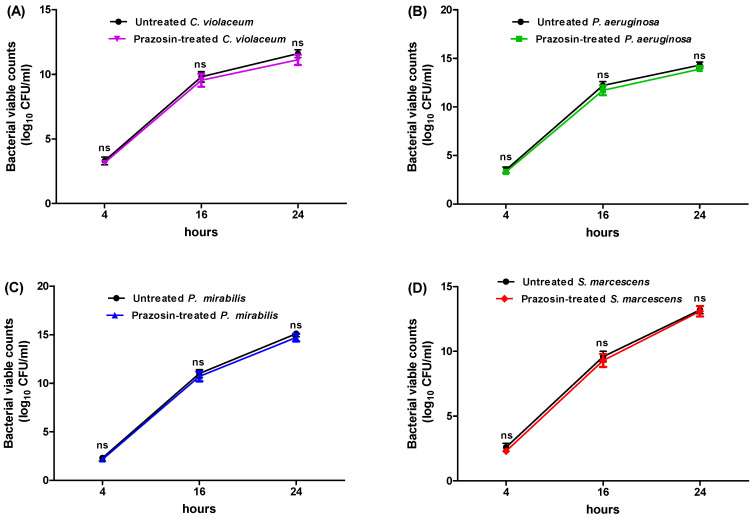
Effect of prazosin on the growth of (**A**) *C. violaceum*, (**B**) *P. aeruginosa*, (**C**) *P. mirabilis*, and (**D**) *S. marcescens*. There were no significant differences between the viable counts of untreated cultures and prazosin-treated cultures. ns: non-significant.

**Figure 2 biology-11-01349-f002:**
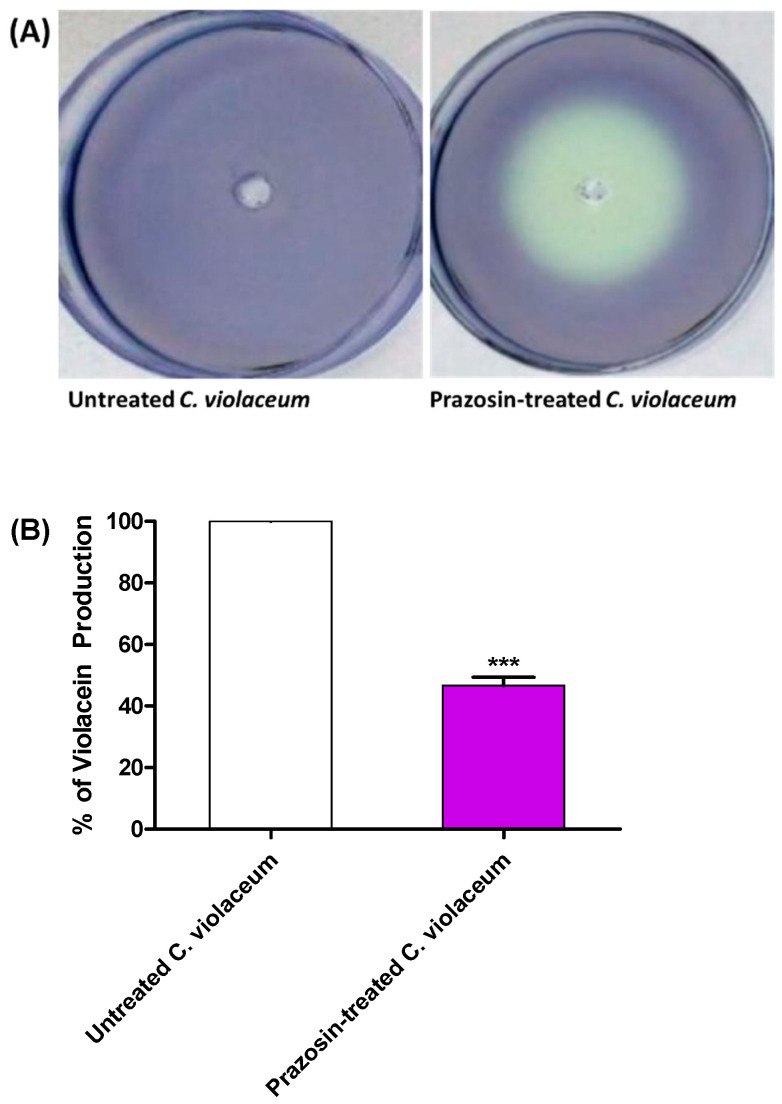
Prazosin decreased the production of *C. violaceum* QS-controlled pigment. (**A**) A white (creamy) zone was formed around the well containing prazosin at sub-MIC, that indicates anti-QS activity. (**B**) The DMSO extracted violacein was quantified in the cultures treated or not with prazosin at sub-MIC. Prazosin significantly diminished the violacein (***: *p* ≤ 0.001).

**Figure 3 biology-11-01349-f003:**
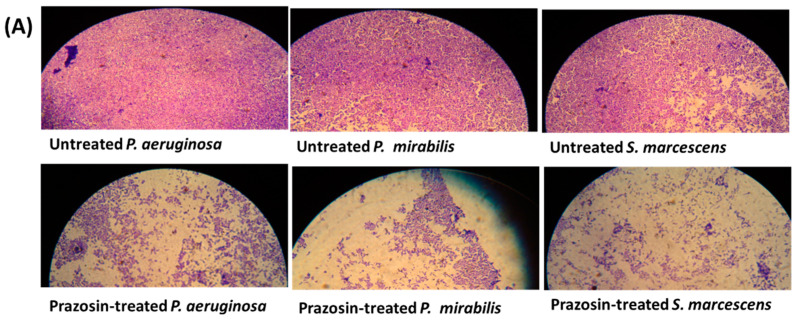
Prazosin diminished the biofilm formation. (**A**) Representative microscopic images showed prazosin effect on diminishing the biofilm formation by *P. aeruginosa*, *P. mirabilis*, and *S. marcescens*. (**B**) The absorbances of crystal violet stained *P. aeruginosa*, *P. mirabilis*, and *S. marcescens* adhered cells. The data are expressed as percentage change from the untreated bacterial strains. Prazosin significantly decreased the biofilm formation (***: *p* ≤ 0.001).

**Figure 4 biology-11-01349-f004:**
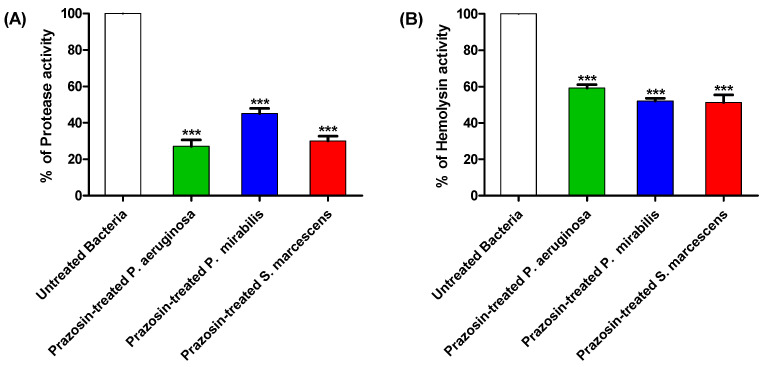
Prazosin decreased the production of virulence factors. Prazosin significantly decreased the production of (**A**) protease, and (**B**) hemolysins (***: *p* ≤ 0.001). The data are expressed as percentage change from the untreated bacterial strains.

**Figure 5 biology-11-01349-f005:**
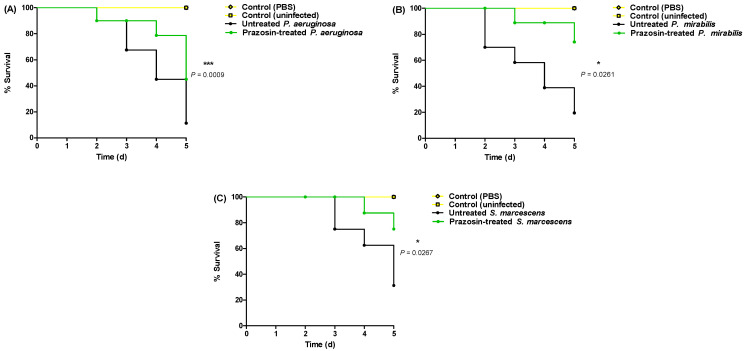
Prazosin protected mice against virulence of (**A**) *P. aeruginosa*, (**B**) *P. mirabilis*, and (**C**) *S. marcescens*. Prazosin showed significant ability to protect the mice from *P. aeruginosa, P. mirabilis, and S. marcescens* pathogenesis (the log rank test for trend *p* = 0.0009, 0.0261, and =0.0267, respectively).

**Figure 6 biology-11-01349-f006:**
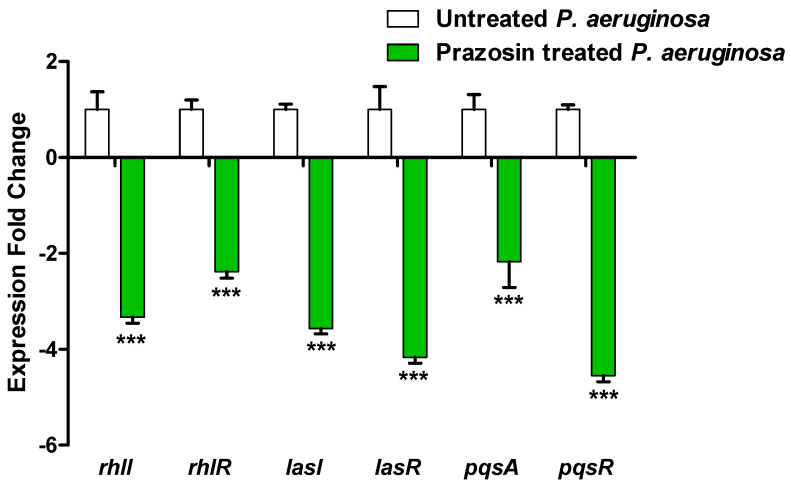
The QS genes of *P. aeruginosa* were less expressed when prazosin was present. From cultures that were either treated or not treated with prazosin at sub-MIC, *P. aeruginosa* RNA was extracted by using RTq-PCR to quantify the expression levels and normalize them to the housekeeping gene *ropD* using the comparative threshold cycle (2^−∆∆Ct^) method. The findings were shown as means ± standard error. Prazosin significantly decreased the expressions of all tested genes, ***: *p* ≤ 0.001.

**Figure 7 biology-11-01349-f007:**
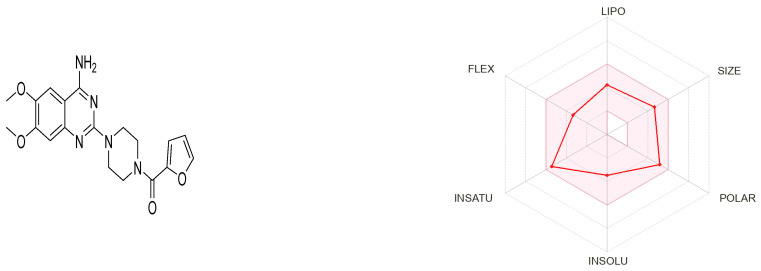
Physicochemical descriptors of prazosin; LIPO (lipophility): −0.7 < XLOGP3 < +5.0, SIZE: 150 g/mol < MV < 500 g/mol, POLAR (polarity): 20 Å^2^ < TPSA < 130 Å^2^, INSOLU (insolubility): −6 < Log S (ESOL) < 0, INSATU (insaturation): 0.25 < Fraction Csp3 < 1, FLEX (flexibility): 0 < Num. rotatable bonds < 9.

**Figure 8 biology-11-01349-f008:**
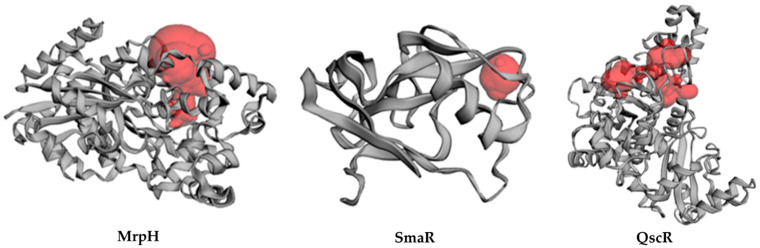
Cartoon representation of the binding site topology at the three bacterial targets, Putative pockets; red color, were calculated via the on-line Computed Atlas of Surface Topography of proteins (CASTp; http://sts.bioe.uic.edu/castp/index.html, accessed on 30 May 2022).

**Figure 10 biology-11-01349-f010:**
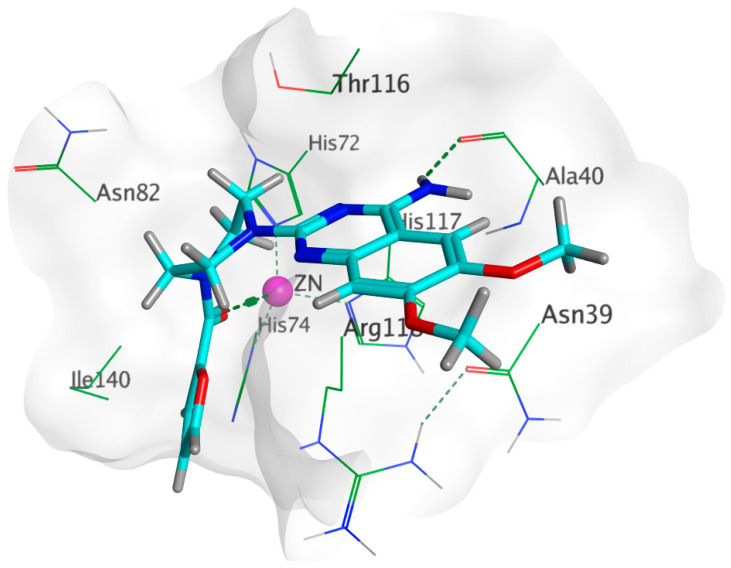
3D binding of prazosin (Cyan) with MrpH active site showing the key amino acid interactions along with the crucial Zn^+2^ binding (purple).

**Figure 11 biology-11-01349-f011:**
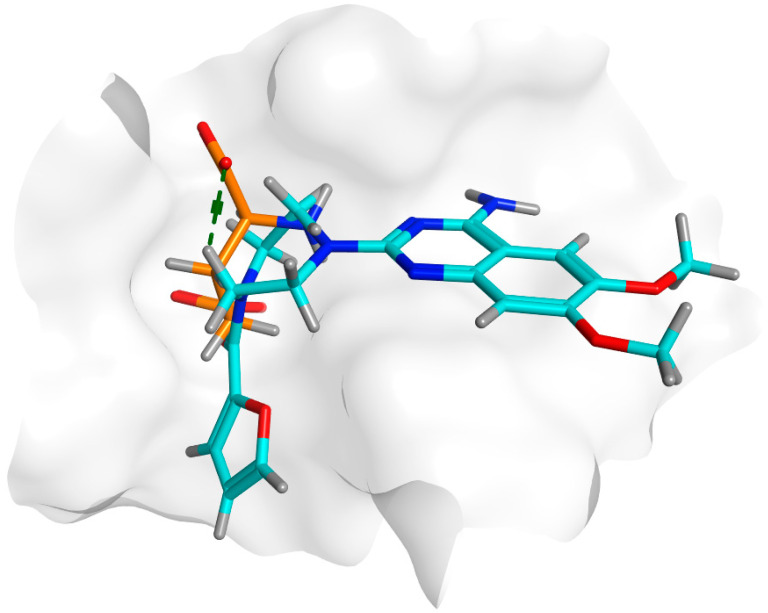
A: 3D alignment of prazosin (cyan) and glutamate (orange) in the binding pocket of MrpH showing the significant difference in size between both.

**Figure 12 biology-11-01349-f012:**
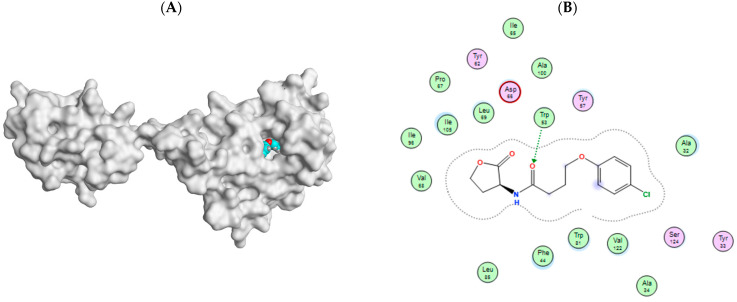
(**A**) SWISS model of *S. marcescens* SmaR with HLC binding site. (**B**) 2D interaction of HLC with *S. marcescens* SmaR for active site validation.

**Figure 13 biology-11-01349-f013:**
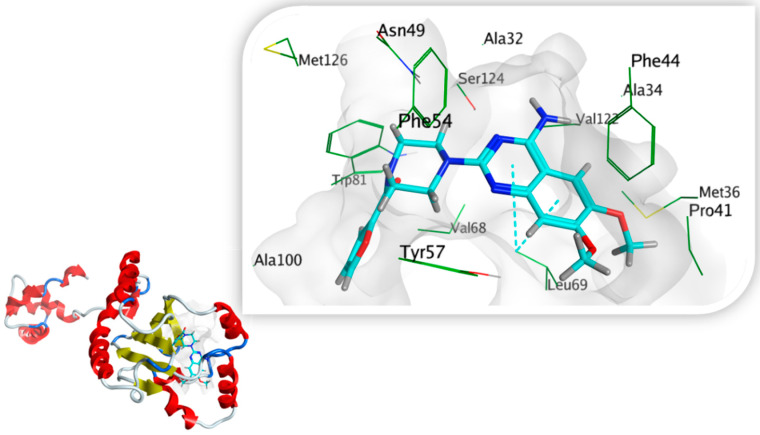
A 3D prazosin-*S. marcescens* SmaR model interaction diagram, prazosin is in cyan thick sticks within the molecular surface of the active site, amino acid residues of the active site are shown as green thin sticks. *Pi-*bond is presented as cyan dots.

**Table 1 biology-11-01349-t001:** Sequences of the used primers in this study.

Target Gene	Sequence (5′–3′)
** *lasI* **	**For:** CTACAGCCTGCAGAACGACA **Rev:** ATCTGGGTCTTGGCATTGAG
** *lasR* **	**For:** ACGCTCAAGTGGAAAATTGG **Rev:** GTAGATGGACGGTTCCCAGA
** *rhlI* **	**For:** CTCTCTGAATCGCTGGAAGG **Rev:** GACGTCCTTGAGCAGGTAGG
** *rhlR* **	**For:** AGGAATGACGGAGGCTTTTT **Rev:** CCCGTAGTTCTGCATCTGGT
** *pqsA* **	**For:** TTCTGTTCCGCCTCGATTTC **Rev:** AGTCGTTCAACGCCAGCAC
** *pqsR* **	**For:** AACCTGGAAATCGACCTGTG **Rev:** TGAAATCGTCGAGCAGTACG
** *rpoD* **	**For:** GGGCGAAGAAGGAAATGGTC **Rev:** CAGGTGGCGTAGGTGGAGAAC

**Table 2 biology-11-01349-t002:** Prazosin molecular descriptors as obtained from PubChem.

Properties	Prazosin
PubChem CID	4893
3D structure	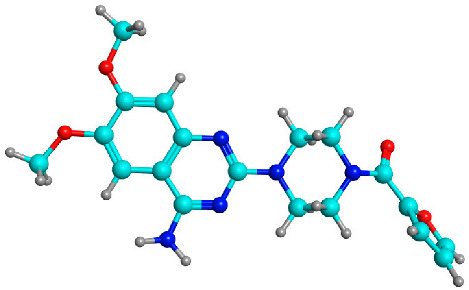
Molecular formula	C_19_H_21_N_5_O_4_
Molecular weight	383.4
LogP	1.3
Hydrogen bond donor	1
Hydrogen bond acceptor	8
Rotatable Bond Count	4
Topological Polar Surface Area (TPSA)	107 Å^2^

**Table 4 biology-11-01349-t004:** Docking results for both glutamate and prazosin with *P. mirabilis* adhesin MrpH.

Ligand	Rigid Receptor Protocol		Induced-Fit Protocol		H-bond Interactions	Hydrophobic Interactions	*pi*-Interactions
	S Score Kcal/mol	RMSD	S Score Kcal/mol	RMSD			
**Prazosin**	**−6.0361**	**1.3998**	−6.0361	1.6343	Ala40. In addition to ionic bond with zinc metal	Asn39, Thr116, Arg118, Ile140	-
**Glutamate**	−8.2383	0.9710	−8.4332	1.4964	Asn 82, Thr116, His117, Arg118. In addition to ionic bond with zinc metal	Asn 82, Thr116, Arg118, Ile140	-

**Table 5 biology-11-01349-t005:** Docking results for both HLC and prazosin with *S. marcescens* SmaR.

Ligand	Rigid Receptor Protocol		Induced-Fit Protocol		H-bond Interactions	Hydrophobic Interactions	*pi*-Interactions
	S Score Kcal/mol	RMSD	S Score Kcal/mol	RMSD			
**Prazosin**	−7.3356	1.6640	−8.0181	1.0449	-	Ala32, Phe44, Tyr57, Leu69, Trp81, Ile105, Val122, Ser124	Leu69 (*pi*-H)
**HLC**	−6.9484	1.3639	−7.1761	1.3923	Trp53	Ala32, Phe44, Tyr57, Trp81, Ile105, Val122, Ser124	-

## Data Availability

All data included in the main text.

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
