# Peer review of "Muting Bacterial Communication: Evaluation of Prazosin Anti-Quorum Sensing Activities against Gram-Negative Bacteria Pseudomonas aeruginosa, Proteus mirabilis, and Serratia marcescens"

_biology, 2022, doi:10.3390/biology11091349_

Round 1
Reviewer 1 Report
This study demonstrated that a compound, prazosin, has significantly decreased the production of violacein and virulent enzymes protease and hemolysins in the tested three strains. The prazosin anti-QS activity was proven by its downregulation of QS-encoding genes and its obvious binding affinity to QS receptors. However, there are several major concerns:
1. Line 52: “Prazosin has significantly diminished the biofilm formation and bacterial virulence in-vivo”. However, biofilm formation assay in vivo was not performed, how did you get the result? For virulence assay in-vivo, there was no repeat, just did once. So, the results are incredible.
2. Line 32: change “….downregulated the QS-encoding genes…” to “….downregulated the expression of QS-encoding genes…”.
3. Lines 131-132: I suggest providing citation.
4. Lines 135-140: anti-biofilm assay, I suggest providing more details on the experimental procedure and using CLSM or SEM to observe the anti-biofilm activity.
5. Line 152: “fresh 2% blood suspensions”, what kind of animals’ blood, horse or rabbit?
6. Lines 159-167: authors just did once repeat, I think bio-assays should be performed at least three times and obtained results could be credible.
7. Section 2.8: why did qRT-PCR assays just for P. aeruginosa not for other two strains?
8. Lines 172-173: “The extracted RNA was used to synthesize cDNA, and RT-PCR was performed to evaluate the QS-encoding genes”, please explain the meanings of RT-PCR and RT-qPCR or qRT-PCR.
9. Line 175: “(∆∆Ct) method”, the correct should be (-∆∆Ct) method or (2-∆∆Ct) method.
10. Please move the contents of section 2.9 to after section 2.10.
11. Lines 209-210: “The prazosin concentrations that inhibited the C. violaceum, P. aeruginosa, P. mirabilis and S. marcescens growth were 2, 2, 1 and 1 mg/mL, respectively.” What kind of concentrations? Is MIC?
12. Figure 1, at 24h, all CFU was greater than 1010, and for (B) and (C), the CFU were about 1015. These results are unbelievable.
13. Lines 239-240 duplicate description with lines 135-136.
14. Figure 5: the authors just did once experiment, how to calculate p value?
15. Figure 6: the presentation is ∆Ct, is incorrect. The correct presentation should be one bar per gene, -∆∆Ct results.
16. Line 445: “P. aeruginosa, P. mirabilis, and S. marcescens.” is not sentence.
17. Lines 468-469: “the bacterial formation to biofilms constitutes an additional obstacle,” the description is wrong.
Reviewer 2 Report
The manuscript showed the benefits of using Prazosin on bacterial infections. Authors demonstrated that Prazosin inhibits quorum sensing in the three gram-negative bacteria Pseudomonas aeruginosa, Proteus mirabilis, and Serratia marcescens and then their virulence, the biofilm formation and the production of extracellular enzymes. Authors showed that Prazosin effects on virulence and quorum sensing did not alter cellular growth and did not induce stress likely to induce resistance. They provided in-vivo experiments and proposed possible mechanisms induced by possible interactions of Prazosin with QS receptors using in-silico docking experiments. The authors concluded that Prazosin could be used as an efficient anti quorum sensing molecule and could be an alternative or a complement to antibiotics, provided that toxicological and pharmacological experiments are carried out.
The manuscript is clear and well written and the results are meaningful. Few technical developments were brought compared to previous works. Indeed, most of the methodologies are mastered and have already been used in previous publications.
Some details below need to be addressed.
1) In the introduction, current uses of Prazosin (e.g medication for high blood pressure, …) were not adequately described.
2) Toxicological studies were not conducted in this work but toxicological effects of Prazosin could be discussed (in the discussion part) given that toxicological analyzes have already been carried out (as the drug is approved). In the light of described effects of Prazosin, does its use seem compatible with a clinical application in term of effects and dose?
3) Concerning the molecular docking experiments, affinities of Prazosin for QS receptors were compared to those obtained with strong ligands (e.g QScR with 3-oxo-C12-HSL or SmaR for HLC). Similar affinities were obtained, except for MrpH, indicating possible interactions. However, negative controls (ligands not associated to QS targets) could be introduced and affinities compared.
4) What is the difference between the molecular docking done between Prazosin and QscR in the publication Almalky et al. (reference 26) and the one made in this work? Why affinities are not similar?
5) RT-PCR showed that mRNA abundances of QS genes were inhibited by Prazosin in P. aeruginosa as demonstrated for terazosin in a precedent work of the laboratory. What about corresponding proteins? Indeed, there is no systematic equivalence in expression levels of a protein and its mRNA. Immunoblots (or proteomics) could be addressed.
6) Abundance levels of autoinducers like AHLs consecutively to Prazosin exposure could also be determined.
7) line 94: α-adrenoreceptors “blockers”.
